# Antipsychotic Chlorpromazine Suppresses STAT5 Signaling, Overcomes Resistance Mediated by the Gatekeeper Mutation FLT3-ITD/F691L, and Synergizes with Quizartinib in FLT3-ITD-Positive Cells

**DOI:** 10.3390/cimb47100797

**Published:** 2025-09-25

**Authors:** Aki Fujii-Hanamoto, Hirokazu Tanaka, Ko Fujimoto, Takahiro Haeno, Yoshiaki Miyake, Ryosuke Fujiwara, Takahiro Kumode, Kentaro Serizawa, Yasuyoshi Morita, Hitoshi Hanamoto, Shinya Rai, Itaru Matsumura

**Affiliations:** 1Department of Hematology and Rheumatology, Kindai University Faculty of Medicine, Osaka-sayama, Osaka 589-8511, Japan; 154300@med.kindai.ac.jp (A.F.-H.); 161926@med.kindai.ac.jp (K.F.); t-haeno182015@med.kindai.ac.jp (T.H.); y.miyake@med.kindai.ac.jp (Y.M.); kumode@med.kindai.ac.jp (T.K.); serizawa@med.kindai.ac.jp (K.S.); moriyasu@med.kindai.ac.jp (Y.M.); rai@med.kindai.ac.jp (S.R.); imatsumura@med.kindai.ac.jp (I.M.); 2Department of Hematology, Kindai University Nara Hospital, Nara 630-0293, Ikoma, Japan; r-fujiwara@med.kindai.ac.jp (R.F.); hanamoto@med.kindai.ac.jp (H.H.); 3Centre for Lymphoid Cancer, BC Cancer, Vancouver, BC V5Z 1L3, Canada

**Keywords:** acute myeloid leukemia, FLT3-ITD, chlorpromazine, STAT5

## Abstract

**Background**: FLT3 mutations, including internal tandem duplication (ITD) and tyrosine kinase domain (TKD) point mutations, represent common genetic alterations in acute myeloid leukemia (AML), with FLT3-ITD associated with poor prognosis. Although FLT3 tyrosine kinase inhibitors (TKIs), such as quizartinib (Quiz) and gilteritinib, have improved clinical outcomes, secondary TKD mutations, particularly the gatekeeper mutation F691L, confer significant resistance. We previously demonstrated that chlorpromazine (CPZ), an antipsychotic drug, inhibits clathrin-mediated endocytosis and selectively suppresses the growth of cancer cells harboring mutant receptor tyrosine kinases. **Methods**: In this study, we examined the efficacy of CPZ in overcoming TKI resistance using Ba/F3 cells expressing FLT3-ITD or FLT3-ITD/F692L, the murine analog of F691L. **Results**: Quiz inhibited proliferation of FLT3-ITD cells but was ineffective against FLT3-ITD/F692L cells. CPZ suppressed growth in both cell types. Co-treatment with CPZ and Quiz exhibited synergistic effects in FLT3-ITD cells, but not in FLT3-ITD/F692L cells. CPZ reduced STAT5 phosphorylation and modulated downstream signaling in FLT3-ITD cells, while only partially affecting STAT5 in FLT3-ITD/F692L cells. Expression of constitutively active STAT5 partially rescued CPZ-induced growth inhibition. **Conclusions**: These findings suggest that STAT5 suppression is a key mechanism of CPZ’s antileukemic activity and support its potential as a therapeutic strategy for FLT3-ITD-positive AML.

## 1. Introduction

FLT3mutations represent one of the most frequently observed genetic alterations in adult acute myeloid leukemia (AML), with internal tandem duplication (FLT3-ITD) mutations detected in approximately 30% of cases [1,2,3]. FLT3-ITD is recognized as an independent adverse prognostic factor, associated with shorter remission duration following initial therapy and higher relapse rates after hematopoietic stem cell transplantation. Particularly poor outcomes have been reported in elderly patients harboring FLT3-ITD mutations. In addition to patients with de novo AML, those with relapsed or refractory FLT3-mutated AML exhibit reduced overall survival and early relapse following conventional chemotherapy [4].

Because FLT3 mutations have long been assumed to play a crucial role in the pathogenesis of AML, the development of FLT3 inhibitors has been actively pursued [2,3,4]. As a result, the first-generation FLT3 tyrosine kinase inhibitor (FLT3 TKI), midostaurin, in combination with chemotherapy, was approved in 2017 for newly diagnosed FLT3-mutated AML in Europe and the United States [5,6]. Subsequently, second-generation FLT3 TKIs, gilteritinib (Gil) and quizartinib (Quiz), were approved as monotherapies for relapsed or refractory AML in Japan [7,8,9,10]. More recently, in 2023, Quiz-based combination therapy was approved for newly diagnosed FLT3-ITD-positive AML in Japan and other countries [11]. The NCCN guidelines now recommend both midostaurin- and Quiz-based regimens as first-line treatments [12].

Compared to first-generation FLT3 TKIs, such as midostaurin, sorafenib, and sunitinib, the second-generation agents Gil and Quiz exhibit greater selectivity and potency against FLT313. Gil, a type I inhibitor, targets both FLT3-ITD and FLT3 tyrosine kinase domain (TKD) mutations (e.g., D835, I836), whereas Quiz, a type II inhibitor, selectively targets FLT3-ITD [7,13]. Due to the selective inhibitory profile of Quiz, TKD mutations are frequently observed in patients who develop resistance after Quiz treatment and are considered a major cause of resistance [14,15]. Among these, the F691L mutation is considered as a gatekeeper mutation and is known to confer resistance to all currently available FLT3 TKIs, posing a significant clinical challenge [16]. FLT3 is a receptor tyrosine kinase consisting of an extracellular ligand-binding domain, a transmembrane region, a juxtamembrane domain, and an intracellular tyrosine kinase domain. FLT3-ITD typically occur in the juxtamembrane domain, leading to constitutive kinase activation, whereas point mutations such as F691L are located within the ATP-binding pocket of the kinase domain. These structural alterations disrupt the binding of type I and type II inhibitors and drive therapeutic resistance [17,18,19]. To provide a visual overview, Appendix A illustrates the domain architecture of FLT3, highlights common ITD and point mutation sites, and depicts representative inhibitor binding modes [17,18,19].

In addition to TKD mutations, a wide range of resistance mechanisms to FLT3 TKIs have been identified. These include overexpression of FLT3 protein, outgrowth of FLT3 mutation-negative clones, and the acquisition of mutations in alternative or downstream signaling pathways such as Ras [13,20]. Furthermore, bone marrow microenvironment-mediated resistance has been reported, involving upregulation of FLT3 ligand and fibroblast growth factor 2 (FGF2) by mesenchymal stromal cells, which activate AML cells via wild-type FLT3 and fibroblast growth factor receptor 1 (FGFR1) [21]. Enhanced expression of CYP3A4, leading to increased metabolic clearance of FLT3 TKIs, has also been implicated [22]. Additionally, persistent STAT5 phosphorylation, AXL upregulation, and enhanced CXCR4 signaling have been shown to contribute to FLT3 TKI resistance [23,24]. Collectively, these findings underscore the need for a comprehensive understanding of resistance mechanisms and the development of novel therapeutic strategies beyond ATP-competitive inhibition to achieve durable remission in FLT3-mutated AML.

Under normal conditions, ligand-bound wild-type receptor tyrosine kinases (RTKs) are incorporated into the cytoplasm through clathrin-dependent endocytosis, transported intracellularly in clathrin-coated vesicles (CCVs), and finally degraded in lysosomes to prevent sustained excessive signaling [25,26,27]. In contrast, mutant RTKs, such as FLT3-ITD and c-KIT V617F found in AML and amplified MET found in gastric, lung, and hepatocellular carcinoma, are reported to evade lysosomal degradation by intracellular transport to endoplasmic reticulum (ER) or Golgi apparatus, and instead continue to propagate oncogenic signaling from these organelles [28,29,30,31,32].

We previously demonstrated that the proliferation of Ba/F3 cells harboring FLT3-ITD or c-Kit V617F was severely impaired by knockdown of clathrin assembly lymphoid myeloid leukemia protein (CALM), which is an essential component of CCVs [33,34]. Consistently, chlorpromazine (CPZ), an antipsychotic that disrupts CCV formation, markedly inhibited proliferation of FLT3-ITD- or c-KIT V617F-positive AML cells, while sparing cells with wild-type FLT3 and c-KIT. Mechanistically, CPZ decreased CALM protein levels and altered the subcellular localization of these mutant RTKs, thereby disrupting its compartment-dependent signaling [34].

In the present study, we investigated whether CPZ alone or in combination with a FLT3 TKI could overcome resistance in FLT3-ITD-expressing Ba/F3 cells harboring the murine F692L mutation, which corresponds to the human F691L gatekeeper mutation. Additionally, we examined the effects of CPZ on key downstream signaling pathways, including MEK/ERK, AKT, and STAT5.

## 2. Materials and Methods

### 2.1. Reagents and Antibodies (Abs)

CPZ and Quiz were purchased from Sigma-Aldrich (St. Louis, MO, USA). A STAT5 inhibitor (Item No. 15784) was purchased from Cayman Chemical (Ann Arbor, MI, USA). These compounds were dissolved in distilled water and stored as 10 mM stock solutions at 4 °C. Recombinant murine IL-3 (rmIL-3) was obtained from PeproTech (Rocky Hill, NJ, USA).

### 2.2. Cell Lines and Cell Cultures

Ba/F3 cells were purchased from the American Type Culture Collection (ATCC, Manassas, VA, USA) and cultured in RPMI 1640 medium (Gibco, Thermo Fisher Scientific, Waltham, MA, USA ) supplemented with 10% fetal calf serum (FCS) and 10 ng/mL rmIL-3. Murine full-length FLT3 wild-type (WT) and FLT3-ITD cDNAs, kindly provided by Dr. Masao Mizuki (Osaka University, Osaka, Japan), were subcloned into a bicistronic retroviral vector, pMSCV-IRES-EGFP. To introduce a gatekeeper mutation (F692L, corresponding to the human F691L mutation) into FLT3-ITD, site-directed mutagenesis was performed using the following primers:Forward: 5′-cat ctc gag cac cat gga tgc ggg cgt tgg-3′Forward (for replacement): 5′-tca cca tag caa caa tat tcc aaa atc aag-3′Reverse (for replacement): 5′-ggc cag tgt act tga ttt tgg aat att gtt-3′Reverse: 5′-cat gtt aac cta act tct ttc tcc gtg aat ctt-3′

A constitutively active form of murine STAT5a (referred to 1*6 STAT5A), harboring S711F and H299R mutations, was generated as previously described [35,36]. The 1*6 STAT5A cDNA was subcloned into the pMSCV-IRES-DsRed vector to construct the expression plasmid.

These retrovirus vectors were transfected into a packaging cell line 293T stably expressing gag and pol. The viral supernatant was harvested 48 h after transfection. Ba/F3 cells were seeded in 3.5 cm dishes coated with a fibronectin fragment (Retronectin Dish, Takara Bio, Shiga, Japan) and cultured with 1 mL of the viral supernatant for 72 h. Infected Ba/F3 cells were then sorted as GFP-positive and/or DsRed-positive populations using a FACSAria cell sorter (BD Biosciences, Franklin Lakes, NJ, USA).

### 2.3. Cell Viability Assays

To examine the effect of CPZ on cell viability, Ba/F3 cells expressing either FLT3-ITD or FLT3-ITD/F692L were seeded into 96-well white plates at a density of 500 cells per well in RPMI 1640 medium supplemented with 10% FCS, and cultured at 37 °C for 72 h. CPZ was tested in a dose range of 0-15 μM for all viability assays. Following incubation with or without CPZ and other indicated drugs, cell viability was assessed using the CellTiter-Glo reagent (Promega, Madison, WI, USA), according to the manufacturer’s instructions. Luminescence was measured with an EnVision plate reader (Wallac 1420 ARVO MX-2, Turku, Finland).

### 2.4. Interactive Analysis and Consensus Interpretation of Multi-Drug Synergies

To analyze response data from multiple drug combinations, we utilized SynergyFinder, version 3.0 (https://synergyfinder.fimm.fi) [37], a web-based application that enables interactive analysis and visualization of drug interaction data. Based on the zero interaction potency (ZIP) model, drug combinations with a peak synergy score exceeding 10 were interpreted as exhibiting significant synergistic effects.

### 2.5. Immunoblotting

Cells were cultured with or without CPZ and the indicated drugs for 36 h. After washing with PBS, cells were lysed using RIPA buffer (Nacalai Tesque, Kyoto, Japan) supplemented with a phosphatase inhibitor cocktail (Nacalai Tesque). Insoluble material was removed by centrifugation.

Equal amounts of protein (15 μg per lane) were subjected to SDS-PAGE using PAGEL gels (Atto, Tokyo, Japan), and subsequently transferred onto polyvinylidene difluoride (PVDF) membranes (Immobilon, Millipore, Bedford, MA, USA) by electrophoresis. Membranes were blocked in TBST blocking buffer (4% non-fat dry milk in Tris-buffered saline with 0.05% Tween 20; 0.15 M NaCl, 0.01 M Tris-HCl, pH 7.4) and then probed with the appropriate primary antibodies (Abs). The following primary Abs were used: anti-Flt3 (#3462), anti-phospho-Flt3 (Tyr589/591, #3464), anti-Erk (#9102), anti-phospho-Erk (Thr202/Tyr204, #4376), anti-Akt (#4691), anti-phospho-Akt (Ser473, #4058), anti-STAT5 (#94205), anti-phospho-STAT5 (Tyr694, #4322), and anti-GAPDH (#5174), all purchased from Cell Signaling Technology (Danvers, MA, USA). The secondary Ab horseradish peroxidase (HRP)-conjugated anti-rabbit IgG was purchased from Promega (W4011). Immune complexes were visualized using an enhanced chemiluminescence detection system (LAS4010, GE Healthcare, Cleveland, OH, USA).

### 2.6. Statistical Analyses

All statistical analyses were performed using EZR software (version 1.54, Saitama Medical Center, Jichi Medical University, Saitama, Japan). At least three independent biological replicates were included in each statistical analysis. Data are presented as mean ± SEM. Comparisons were performed using one-way analysis of variance (ANOVA) followed by Dunnett’s post hoc test, repeated-measures ANOVA, or Welch’s ANOVA, as appropriate. A *p*-value of <0.05 was considered statistically significant. Flow cytometry data were analyzed using the FlowJo software package (version 10.7.1; Ashland, OR, USA).

## 3. Results

### 3.1. Effects of CPZ and Quiz on FLT3-ITD-Dependent Proliferation of Ba/F3 Cells

We first transfected FLT3-ITD and FLT3-ITD carrying the gatekeeper mutation F692L (FLT3-ITD/F692L) into the murine IL-3-dependent Ba/F3 cell line. After transfection, both cell clones acquired IL-3-independent growth (Figure 1A, CTL). When Quiz was added in the absence of IL-3, proliferation of FLT3-ITD-transfected cells was inhibited in a concentration-dependent manner, with complete suppression observed at concentrations of 1.0 nM or higher (Figure 1A, left panel; IC_50_ = 0.68 nM). In contrast, proliferation of FLT3-ITD/F692L-transfected cells was not inhibited by Quiz, even at concentrations of 1.0 nM or higher (Figure 1A, right panel), indicating that these cells are resistant to FLT3 inhibition.

Under the same culture conditions, CPZ inhibited proliferation in FLT3-ITD-transfected cells in a concentration-dependent manner, consistent with our previous report (Figure 1B, left panel; IC_50_ = 10.34 μM). Similarly, proliferation of FLT3-ITD/F692L-transfected cells was also suppressed in a concentration-dependent manner (Figure 1B, right panel; IC_50_ = 9.60 μM). No significant difference in the growth-inhibitory effect was observed between the two cell lines.

### 3.2. Synergistic Inhibitory Effect of CPZ with Quiz in FLT3-ITD-Expressing Cells

We next evaluated the combined effect of CPZ and Quiz on the proliferation of FLT3-ITD-expressing and FLT3-ITD/F692L-expressing cells using an ATP assay. Both drugs were tested at various concentrations in combinations: up to 1.5 nM for Quiz, at which it alone exhibited near-complete inhibitory activity (Figure 1A, left panel), and up to 15 μM for CPZ, at which it almost completely suppressed the growth of FLT3-ITD- expressing cells (Figure 1B, left panel).

The 72 h growth curves of FLT3-ITD-expressing cells treated with Quiz and/or CPZ are shown in Figure 2A (upper panel). Synergistic effects were analyzed using SynergyFinder, and the results are presented in Figure 2B (left panel). The combination of CPZ and Quiz demonstrated a marked synergistic effect in FLT3-ITD-expressing cells, with a maximum ZIP synergy score of 18.7 observed in the concentration range of 5–10 μM CPZ and 0.2–0.6 nM Quiz. These findings indicate that a significant synergistic growth-inhibitory effect was achieved at lower concentrations of each drug.

The same drug combination was tested in FLT3-ITD/F692L-expressing cells under identical conditions. The corresponding growth curves are shown in Figure 2A (lower panel), and the synergy analysis results are presented in Figure 2B (right panel). In this cell line, only a slight synergistic effect was observed at lower concentrations of CPZ (0–1 μM) and Quiz (0.1–0.4 nM), with a maximum ZIP score of 2.72, which was not considered statistically significant. These results suggest that CPZ neither restored Quiz sensitivity in the Quiz-resistant FLT3-ITD/F692L-expressing cells, nor produced an additive inhibitory effect.

### 3.3. Effects of CPZ on AKT and STAT5 Activities

We next examined the effects of CPZ on the activation of signaling molecules downstream of FLT3 by Western blotting in FLT3-ITD- and FLT3-ITD/F692L-transfected cell lines. Cells were treated with CPZ at 0, 5, 7, and 10 μM for 2 h under culture conditions containing 10% FCS in the absence of IL-3. Figure 3A,B show the expression levels of total and phosphorylated (activated) AKT, ERK, and STAT5 in FLT3-ITD- and FLT3-ITD/F692L-transfected cell lines, respectively.

Quantitative analyses showing the ratio of phosphorylated to total protein levels are presented on the right panels of Figure 3. In FLT3-ITD-transfected cells, phosphorylation of FLT3 was suppressed by CPZ in a dose-dependent manner. However, total FLT3 protein levels also decreased with increasing CPZ concentrations, resulting in a relative increase in the phosphorylation ratio. This change, however, was not statistically significant compared to the control (0 μM CPZ). In contrast, phosphorylation of AKT and ERK increased in a dose-dependent manner, with no notable changes in their total protein levels. Significant activation of both AKT and ERK was observed at CPZ concentrations of 5 μM or higher, with near-maximal activation reached at 7 μM. Conversely, CPZ treatment did not affect total STAT5 protein levels but caused a dose-dependent suppression of STAT5 phosphorylation. Significant inhibition of STAT5 activation was observed at CPZ concentrations of 5 μM or more compared to the control.

A similar analysis was conducted in FLT3-ITD/F692L-transfected cells. In this mutant line, CPZ showed scarce effect on FLT3 phosphorylation, while it induced AKT activation. Although CPZ significantly suppressed STAT5 phosphorylation, its inhibitions needed higher concentration (10 μM) of CPZ compared to that observed in FLT3-ITD–transfected cells. In contrast to FLT3-ITD-transfected cells, CPZ had no apparent effect on ERK phosphorylation in FLT3-ITD/F692L-transfected cells.

### 3.4. Effect of Constitutively Active STAT5 on CPZ-Mediated Growth Inhibition in FLT3-ITD-Expressing Cells

To elucidate the significance of STAT5 activity reduction induced by CPZ in FLT3-ITD-expressing cells, we examined the effect of forced expression of a constitutively active form of STAT5A (1*6 STAT5A) on CPZ-mediated growth inhibition in FLT3-ITD-transfected cells. Ba/F3 cells previously transfected with FLT3-ITD were further transfected with either an empty vector (Mock) or a 1*6 STAT5A expression vector. To assess changes in STAT5 activity, each cell line was cultured in the absence of FCS and IL-3 for 8 h, followed by treatment with 10 μM CPZ for 2 h. STAT5 protein expression and phosphorylation levels were then analyzed by Western blotting. In Mock-transfected cells, STAT5 phosphorylation was significantly suppressed by CPZ treatment. In contrast, 1*6 STAT5A-transfected cells exhibited higher baseline STAT5 activity than Mock cells, and although CPZ treatment suppressed its activity, the decrease was not statistically significant (Figure 4A).

The effect of CPZ on the proliferation of each cell line was subsequently evaluated by an ATP assay after 72 h treatment. In Mock-transfected cells, proliferation was inhibited in a dose-dependent manner at CPZ concentrations of 5 μM or higher. In contrast, 1*6 STAT5A-transfected cells exhibited no significant growth inhibition at CPZ concentrations up to 7 μM, and only modest, dose-dependent inhibition at concentrations above 10 μM. Overall, the CPZ-mediated growth-inhibitory effect was significantly attenuated in 1*6 STAT5A-transfected cells compared to Mock-transfected control cells (Figure 4B). These results suggest that the growth-inhibitory effect of CPZ on FLT3-ITD-expressing cells is, at least partly, mediated through the suppression of STAT5 activity.

### 3.5. Additive Growth Inhibitory Effect of CPZ and a STAT5 Inhibitor in FLT3-ITD-Expressing and FLT3-ITD/F692L-Expressing Cells

As CPZ did not completely inhibit STAT5 activity, and its effects were considered to involve mechanisms other than STAT5 suppression, we next examined the combinatory effect of CPZ and a STAT5 inhibitor. Each cell line was cultured in medium containing 10% FCS and lacking IL-3 for 2 h in the presence of 100 nM STAT5 inhibitor and/or 10 μM CPZ. STAT5 protein expression and phosphorylation levels were analyzed by Western blotting. In FLT3-ITD-transfected cells, STAT5 phosphorylation was completely suppressed by the STAT5 inhibitor alone, as well as in combination with CPZ (Figure 5A, left panel). Total STAT5 protein levels showed a decreasing trend upon treatment but did not change significantly. In FLT3-ITD/F692L-transfected cells, STAT5 phosphorylation was partially suppressed by the STAT5 inhibitor alone and was completely inhibited by the combination with CPZ (Figure 5A, right panel). Neither treatment significantly affected total STAT5 protein levels in these cells.

To evaluate the impact on proliferation, ATP assays were performed after 72 h of treatment under the same conditions. CPZ reduced the proliferation of FLT3-ITD-transfected cells by approximately 50%, and that of FLT3-ITD/F692L-transfected cells by about 60%. The STAT5 inhibitor alone reduced cell proliferation to approximately 20% of the control level in both cell lines. Importantly, the combination of the STAT5 inhibitor and CPZ resulted in a more pronounced suppression of proliferation in both cell lines (Figure 5B).

We also evaluated the potential synergistic effects of CPZ (0–15 μM) in combination with the STAT5 inhibitor (0–500 nM) using the SynergyFinder platform. In FLT3-ITD–transfected cells, the combination of the two agents induced growth inhibition at lower concentrations. A slight, but not statistically significant, synergistic effect was observed, with a maximum ZIP synergy score of 7.75 within the range of 5–10 μM CPZ and 10–100 nM STAT5 inhibitor (Figure 5C, left panel). Similarly, in FLT3-ITD/F692L-transfected cells, no statistically significant synergistic effect was detected (Figure 5C, right panel). Nevertheless, the combination exhibited an additive inhibitory effect on cell proliferation compared to each agent alone.

## 4. Discussion

FLT3-ITD mutations are associated with a poor prognosis in AML and remain a key therapeutic target. Although the development of FLT3 TKIs, including midostaurin, Quiz and Gil, has improved clinical outcomes, resistance remains a major clinical challenge. In particular, the gatekeeper mutation F691L, which confers resistance to both type I and type II FLT3 inhibitors, poses a significant barrier to achieving durable remission [14,15,38].

We previously found that CPZ selectively targets AML cells and AML stem cells harboring mutant RTKs by altering their intracellular trafficking and subcellular localization [29]. In addition, we demonstrated that CPZ inhibited the growth/survival of the non-small cell lung cancer cell line, PC9 harboring an activating mutation (exon 19 deletion) in the epidermal growth factor receptor (EGFR) gene [39]. Notably, CPZ not only suppressed the growth/survival of gefitinib (GEF)-resistant PC9ZD cells with gatekeeper mutation T790M, but also restored their sensitivities to GEF. Given that CPZ exerts its effects through a mechanism completely different from ATP-competitive FLT3 TKIs, we hypothesized that CPZ might be effective against the FLT3-ITD/F691L mutation.

In the present study, we investigated the antileukemic effects of CPZ in FLT3-ITD-positive Ba/F3 cells, including those harboring the F692L mutation (murine homolog of human F691L). Our results demonstrated that CPZ inhibited the proliferation of both FLT3-ITD and FLT3-ITD/F692L-expressing Ba/F3 cells. CPZ and Quiz exerted a synergistic growth-inhibitory effect in FLT3-ITD-expressing cells, but not in FLT3-ITD/F692L-expressing cells, indicating that the F692L mutation abrogates Quiz sensitivity but does not affect CPZ efficacy.

Mechanistically, CPZ reduced STAT5 phosphorylation and induced activation of AKT and ERK in FLT3-ITD-expressing cells, whereas ERK activation was not observed in F692L-mutant cells. This observation suggests that FLT3-ITD/F692L cells may bypass ERK-dependent signaling and instead rely more heavily on alternative pathways such as AKT signaling. Previous studies have reported that the analogous human F691L mutation can alter downstream signaling and promote resistance through enhanced AKT activation and microenvironment-mediated survival mechanisms. Notably, the enforced expression of constitutively active STAT5 partially attenuated CPZ-induced growth inhibition. In addition, CPZ revealed additive, though not synergistic, effects when combined with a selective STAT5 inhibitor in both cell lines. The additive, rather than synergistic, effect is likely because the STAT5 inhibitor was used at a concentration sufficient to fully block STAT5 phosphorylation, leaving little room for further suppression by CPZ. This outcome suggests that CPZ exerts its effect via STAT5-independent pathway(s), consistent with its pleiotropic actions.

These findings indicate that suppression of STAT5 signaling is, at least partly, involved in mechanism by which CPZ exerts its antileukemic effects. In contrast to the proliferation assays, we did not observe a consistent dose-dependent changes in phosphorylation for FLT3, or other signaling proteins in Western blot analyses. This discrepancy may reflect the transient and reversible nature of phosphorylation as well as compensatory signaling mechanisms that modulate kinase activity over time. Nevertheless, significant inhibition of STAT5 phosphorylation was detected at concentrations within the clinically relevant plasma range of CPZ (1–20 μM), underscoring the potential therapeutic applicability of our findings. To clarify the significance of Akt activation in CPZ-induced growth suppression of FLT3-ITD-expressing cells, we treated the cells with CPZ alone or in combination with the Akt inhibitor capivasertib. As a result, capivasertib further enhanced CPZ-induced growth inhibition, suggesting that Akt activation represents a compensatory response rather than contributing to CPZ-mediated growth suppression. These findings indicate that CPZ-induced activation of AKT, and to a lesser extent ERK, may represent compensatory signaling that counteracts its growth-inhibitory effects. Therefore, combining CPZ with inhibitors targeting these pathways, such as AKT inhibitors, could further enhance therapeutic efficacy. Such combination approaches warrant further investigation as potential strategies to overcome compensatory signaling and optimize the clinical benefit of CPZ.

We previously reported that CPZ inhibits the proliferation of AML and lung adenocarcinoma cells harboring activating RTK mutations by disrupting CCV formation and altering the subcellular localization of RTKs such as FLT3-ITD and c-Kit V617F [33,34]. However, CPZ also inhibited growth of AML cells without RTK mutations, albeit to a lesser extent. This suggests that CPZ’s antitumor activity is not solely dependent on RTK mutation status and that CPZ likely engages multiple mechanisms of action. In support of this, recent studies have demonstrated that CPZ induces G2/M cell cycle arrest [40,41], promotes apoptosis via mitochondrial and lysosomal pathways [42], modulates autophagy, increases reactive oxygen species (ROS) production [43,44], and inhibits P-glycoprotein (P-gp)-mediated drug efflux in various cancer types [45,46]. These observations suggest that CPZ exerts pleiotropic effects on cancer cells and may overcome resistance mechanisms related to ATP-competitive inhibition.

Specifically, the ability of CPZ to retain efficacy in cells harboring the FLT3-ITD/F692L mutation highlights its potential as a therapeutic option in cases where conventional FLT3 inhibitors fail. The additive effects of CPZ with STAT5 inhibition further support its role in combination strategies aimed at enhancing efficacy and potentially preventing resistance. In addition, it will be important for future studies to investigate CPZ in combination with other standard-of-care AML treatments, such as cytarabine, daunorubicin, azacitidine, and venetoclax to further assess its translational potential. Although our findings suggest that CPZ interferes with FLT3 signaling, a direct molecular interaction between CPZ and FLT3 has not been established. Biophysical methods such as isothermal titration calorimetry (ITC) or fluorescence resonance energy transfer (FRET) assays would be valuable for determining whether CPZ binds directly to FLT3 or exerts its effects through indirect mechanisms. Future studies employing these approaches will help elucidate the molecular basis of CPZ activity and further strengthen the rationale for its clinical repurposing. A limitation of our study is that all experiments were performed using Ba/F3 cell models. While these systems are widely used to investigate the oncogenic potential of FLT3 mutations, they do not fully recapitulate the complexity of human AML. Moreover, as CPZ is a repurposed psychiatric drug with known CNS effects, its efficacy and safety must be validated in in vivo AML models before clinical application. Such studies will be essential to establish appropriate dosing strategies that maximize antileukemic activity while minimizing neurological side effects.

## 5. Conclusions

Collectively, our findings suggest that CPZ may represent a promising therapeutic agent for the treatment of FLT3-ITD-positive AML, especially for the patients with TKI-resistant mutations. Future studies are warranted to further elucidate the downstream signaling networks modulated by CPZ and to evaluate its in vivo efficacy and potential synergy with other agents in clinically relevant AML models. Authors should discuss the results and how they can be interpreted from the perspective of previous studies and of the working hypotheses. The findings and their implications should be discussed in the broadest context possible. Future research directions may also be highlighted.

## Figures and Tables

**Figure 1 cimb-47-00797-f001:**
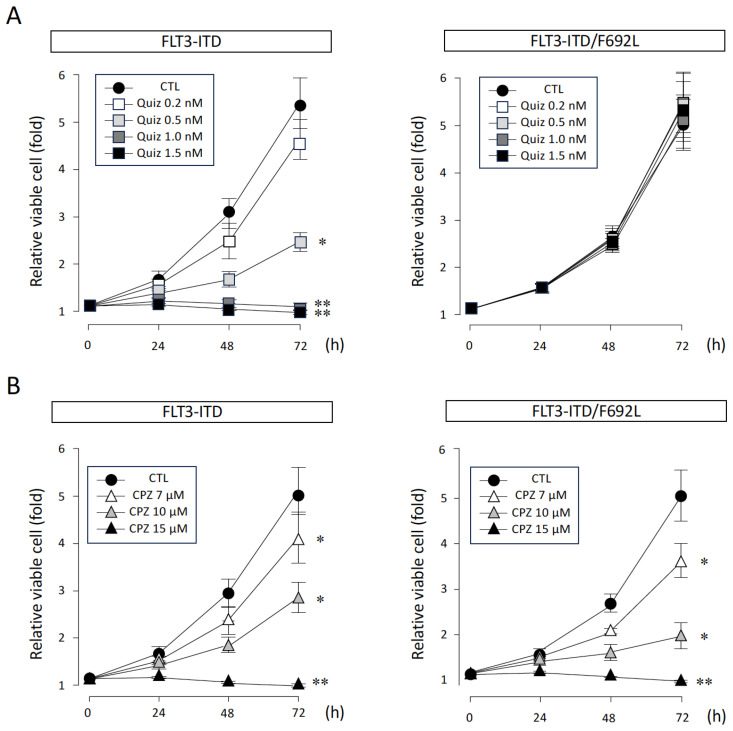
Effects of CPZ on the growth and survival of FLT3-ITD- and FLT3-ITD/F692L-transfected Ba/F3 cells. (**A**) Growth inhibition by quizartinib (Quiz) was assessed using an ATP assay following treatment with 0 (control, CTL), 0.2, 0.5, 1.0, and 1.5 nM of Quiz. (**B**) Growth inhibition by CPZ was evaluated following treatment with 0 (CTL), 7, 10, and 15 μM of CPZ using the same assay. Ba/F3 cells expressing either FLT3-ITD or FLT3-ITD/F692L were seeded at 2000 cells per well in 96-well white plates and cultured in RPMI 1640 medium supplemented with 10% FCS at 37 °C for 72 h. Cell viability was measured using the Cell Titer-Glo reagent (Promega) according to the manufacturer’s instructions. Relative proliferation was calculated by normalizing to the value at day 0 (set as 1). Data represent the mean ± SEM of three independent experiments. Statistical significance was determined by two-sided unpaired Student’s *t*-test; * *p* < 0.05, ** *p* < 0.01.

**Figure 2 cimb-47-00797-f002:**
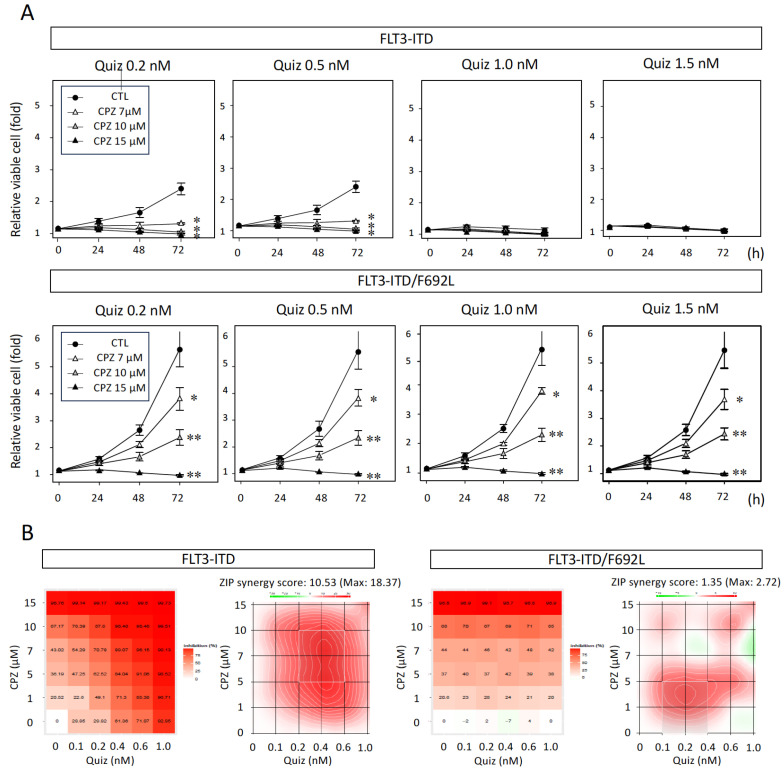
Effects of CPZ in combination with Quiz on the proliferation of FLT3-ITD- or FLT3-ITD/F692L-transfected cells. (**A**) Ba/F3 cells expressing either FLT3-ITD or FLT3-ITD/F692L were seeded at 2,000 cells per well in 96-well white plates and cultured in RPMI 1640 medium supplemented with 10% FCS at 37 °C for 72 h. The effects of Quiz (0.2, 0.5, 1.0, and 1.5 nM) and CPZ (0 [control, CTL], 7, 10, and 15 μM) on cell proliferation were evaluated using an ATP assay. Upper panel: FLT3-ITD-transfected cells; lower panel: FLT3-ITD/F692L-transfected cells. Relative proliferation was calculated by normalizing to the value at day 0 (set as 1). Data are shown as mean ± SEM from three independent experiments. Statistical significance was assessed by two-sided unpaired Student’s *t*-test; * *p* < 0.05, ** *p* < 0.01. (**B**) Synergistic effects of CPZ and Quiz were evaluated based on ATP assay data using the SynergyFinder Plus platform. Cells were treated with Quiz (0, 0.1, 0.2, 0.4, 0.6, and 1.0 nM) in combination with CPZ (0, 1, 5, 7, 10, and 15 μM). Left panel: FLT3-ITD-transfected cells; right panel: FLT3-ITD/F692L-transfected cells. The dose–response matrix shows the percent growth inhibition (left figures), and synergy scores are visualized using ZIP models (right figures). Red regions indicate synergy (ZIP synergy score > 0), and green regions indicate antagonism (ZIP synergy score < 0).

**Figure 3 cimb-47-00797-f003:**
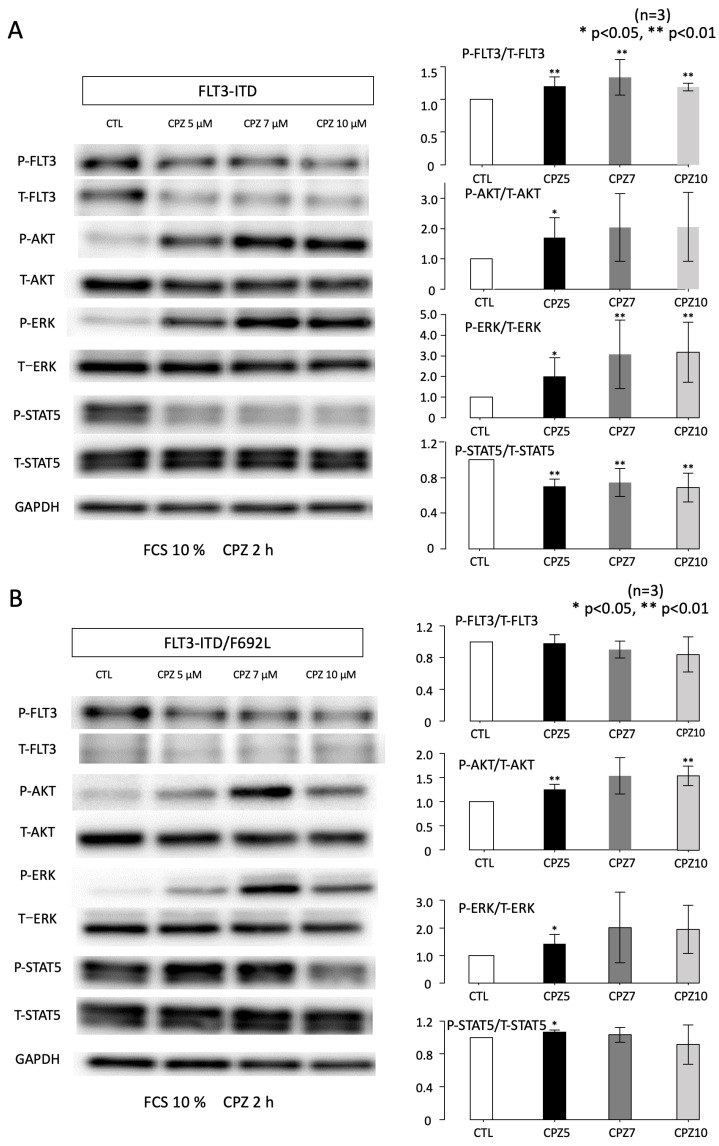
Altered activation of downstream signaling molecules upon CPZ treatment in FLT3-ITD- and FLT3-ITD/F692L-transfected cells. Ba/F3 cells transfected with either FLT3-ITD (**A**) or FLT3-ITD/F692L (**B**) were treated with various concentrations of chlorpromazine (CPZ) for 2 h. Whole-cell lysates were prepared and subjected to immunoblotting using the indicated antibodies. Densitometric analyses were performed with Image Quant TL software (Ver 10.2)using data from two independent experiments. The signal intensities of phosphorylated proteins were normalized to the corresponding total protein levels and are shown as dot plots. Data are presented as mean ± SEM. Statistical comparisons were made using two-sided unpaired Student’s *t*-tests.

**Figure 4 cimb-47-00797-f004:**
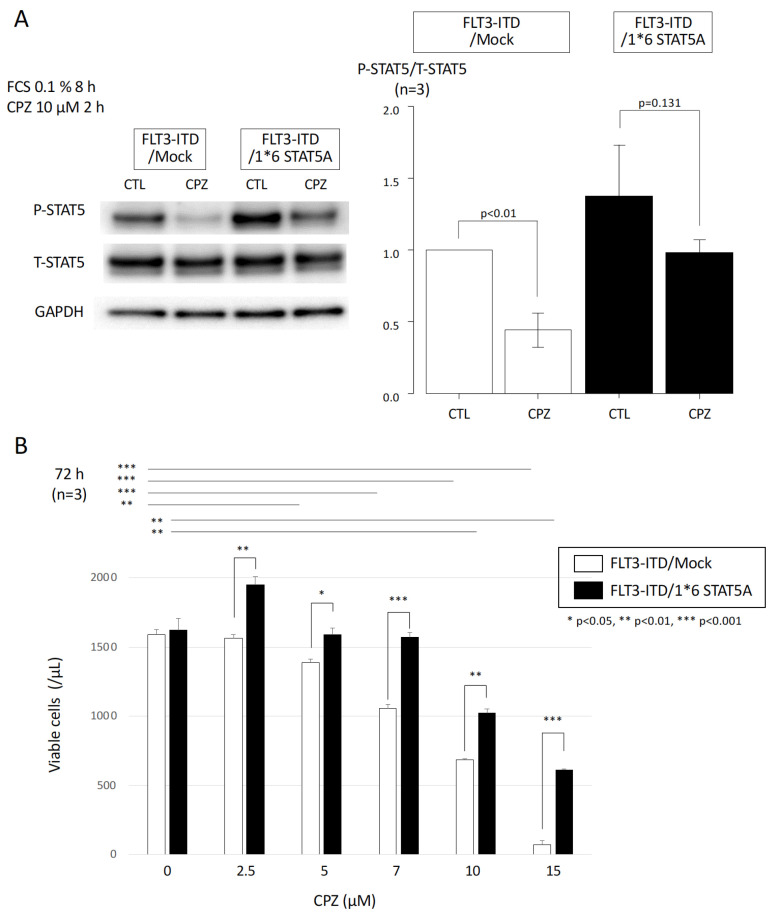
Significance of altered STAT5 activity in FLT3-ITD-transfected cells upon CPZ treatment. (**A**) Ba/F3 cells transfected with FLT3-ITD were further transfected with either an empty vector (Mock) or a constitutively active form of STAT5A (1*6 STAT5A). Cells were cultured in the absence of FCS and IL-3 for 8 h, followed by treatment with 10 μM CPZ for 2 h. Whole-cell lysates were prepared and subjected to immunoblotting using the indicated antibodies. Densitometric analysis was performed using Image Quant TL software with data from three independent experiments. Dot plots represent the ratio of phosphorylated to total protein levels. Data are presented as mean ± SEM. Statistical comparisons were made using two-sided unpaired Student’s *t*-tests. (**B**) Effects of CPZ on cell proliferation were assessed by ATP assay after 72 h of treatment. Ba/F3 cells expressing either FLT3-ITD/Mock or FLT3-ITD/1*6 STAT5A were seeded at 2000 cells per well in 96-well white plates and cultured in RPMI 1640 medium supplemented with 10% FCS at 37 °C. Cell viability was measured using the Cell Titer-Glo reagent (Promega) according to the manufacturer’s instructions. Relative proliferation was normalized to the value at day 0 (set as 1). Data are shown as mean ± SEM from three independent experiments. Statistical significance was determined using a two-sided unpaired Student’s *t*-test: * *p* < 0.05, ** *p* < 0.01, *** *p* < 0.001.

**Figure 5 cimb-47-00797-f005:**
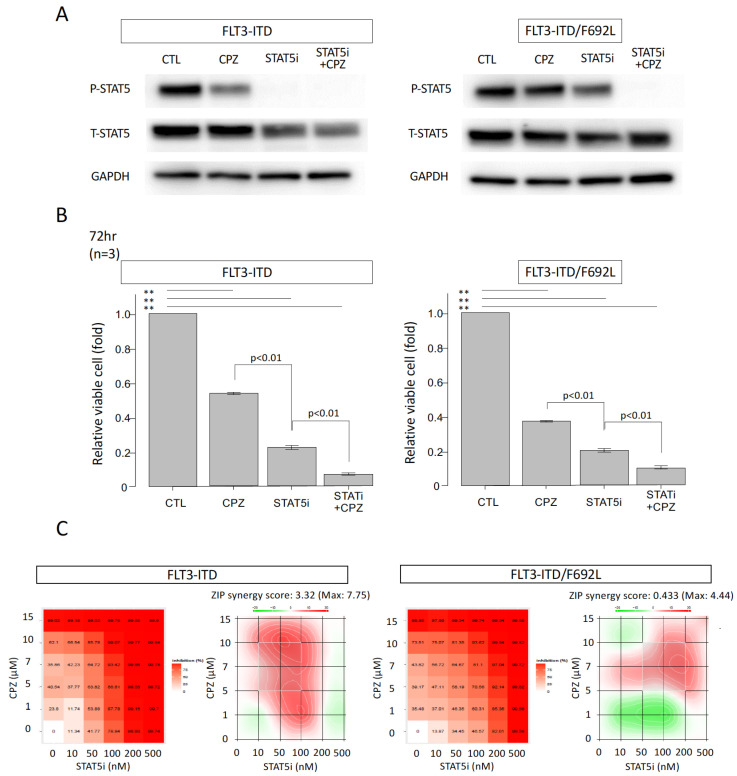
Effects of CPZ in combination with a STAT5 inhibitor on the proliferation of FLT3-ITD- and FLT3-ITD/F692L-transfected cells. (**A**) Ba/F3 cells transfected with either FLT3-ITD or FLT3-ITD/F692L were treated for 2 h with 100 nM of a STAT5 inhibitor and/or 10 μM CPZ. Whole-cell lysates were prepared and subjected to immunoblotting using the indicated antibodies. (**B**) Effects of CPZ in combination with a STAT5 inhibitor on cell proliferation were assessed by ATP assay. Ba/F3 cells expressing either FLT3-ITD or FLT3-ITD/F692L were treated with 100 nM STAT5 inhibitor and/or 10 μM CPZ and cultured for 72 h. Viable cell numbers were measured using the Cell Titer-Glo reagent (Promega) according to the manufacturer’s instructions. Relative proliferation was normalized to the untreated control (CTL), which was set as 1. Data are presented as mean ± SEM from three independent experiments. Statistical significance was determined by two-sided unpaired Student’s *t*-test; ** *p* < 0.01. (**C**) Synergistic effects of the STAT5 inhibitor and CPZ were evaluated based on ATP assay data. Cells were treated with varying concentrations of STAT5 inhibitor (0, 10, 50, 100, 200, and 500 nM) and CPZ (0, 1, 5, 7, 10, and 15 μM). Synergy analysis was performed using SynergyFinder Plus, and results are presented for FLT3-ITD-transfected cells (left panel) and FLT3-ITD/F692L-transfected cells (right panel). The dose–response matrix plots (left figures) represent percent growth inhibition, while ZIP synergy distribution maps (right figures) show synergy scores: red indicates synergy (ZIP score > 0), and green indicates antagonism (ZIP score < 0).

## Data Availability

The data presented in this study are available upon request from the corresponding author.

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
