# Peer review of "Antipsychotic Chlorpromazine Suppresses STAT5 Signaling, Overcomes Resistance Mediated by the Gatekeeper Mutation FLT3-ITD/F691L, and Synergizes with Quizartinib in FLT3-ITD-Positive Cells"

_cimb, 2025, doi:10.3390/cimb47100797_

Round 1
Reviewer 1 Report
Comments and Suggestions for Authors
This manuscript presents a robust and well-structured investigation into the therapeutic potential of the antipsychotic drug chlorpromazine (CPZ) in FLT3-ITD-positive acute myeloid leukemia (AML), including cells resistant to tyrosine kinase inhibitors (TKIs) due to the gatekeeper F691L mutation. The findings are novel and clinically relevant, addressing a critical problem in AML treatment: TKI resistance.
The study combines pharmacological, molecular, and cell biology approaches to demonstrate that CPZ inhibits cell proliferation by suppressing STAT5 signaling and shows synergy with quizartinib in sensitive cells. The experiments are well-designed, controls are appropriate, and the conclusions are supported by the data.
- The study does not provide a clear dose-response relationship for all observed effects of CPZ, particularly in the Western blot analyses. This makes it difficult to determine the optimal dosage for therapeutic effects while minimizing potential side effects.
- The study shows differences in the effects of CPZ on signaling pathways between FLT3-ITD and FLT3-ITD/F692L cell lines, particularly in ERK activation. These inconsistencies are not fully explained, which could indicate a more complex mechanism of action than initially proposed
- While the study examines the combination of CPZ with quizartinib and a STAT5 inhibitor, it does not explore combinations with other standard-of-care treatments for AML. This limits the translational potential of the findings.
- CPZ is known to have multiple effects on cellular processes beyond its intended target. The study does not thoroughly address potential off-target effects that could contribute to the observed anti-leukemic activity. This limitation makes it difficult to attribute the effects solely to FLT3-ITD inhibition.
- Effectively shows the strong synergy in FLT3-ITD cells and its absence in FLT3-ITD/F692L cells. The use of the ZIP model is appropriate. (figure 2)
- The Western blot data are clear. The quantitative analysis would benefit from error bars (SEM) and statistical analysis to confirm the stated significances. (figure 3)
- The additive effect with the STAT5 inhibitor is interesting. The authors should clarify whether the STAT5 inhibitor was used at a concentration that fully inhibits STAT5 phosphorylation (which the blot in 5A suggests it does) and discuss why the effect is only additive and not synergistic. (figure 5)
Author Response
We sincerely thank the reviewer for the thoughtful and constructive comments. We are encouraged by the recognition of the novelty and robustness of our study. Below, we provide point-by-point responses and describe the corresponding revisions made in the manuscript.
Comment 1.
The study does not provide a clear dose-response relationship for all observed effects of CPZ, particularly in the Western blot analyses. This makes it difficult to determine the optimal dosage for therapeutic effects while minimizing potential side effects.
Response:
We appreciate the reviewer’s comment. We previously reported that CPZ inhibits proliferation of Ba/F3 cells expressing FLT3-ITD in a dose-dependent manner (Rai S, et al. Nature Communications 2020). In this assay, ICâ‚…â‚€ was 6.94 μM, and this value was used to determine the CPZ concentration employed in the present experiments. As a result, CPZ inhibited cell growth in a dose-dependent manner in the current assay (ICâ‚…â‚€ 10.34 μM). However, in Western blot analyses, we did not observe a consistent dose-dependent changes of phosphorylation for FLT3, or downstream proteins. This discrepancy may reflect the dynamic and transient nature of phosphorylation events as well as compensatory signaling feedback mechanisms. To address the reviewer’s concern, we have now added SEM values and statistical analyses to the quantitative Western blot data. Importantly, we emphasize that significant inhibition of STAT5 phosphorylation was achieved at CPZ concentrations within the clinically achievable plasma range (1-20 μM), supporting the translational relevance of our findings. We have revised the Discussion to explain this point (line 497~504 in revision).
In contrast to the proliferation assays, we did not observe a consistent dose-dependent changes of phosphorylation for FLT3, or other signaling proteins in Western blot analyses. This discrepancy may reflect the transient and reversible nature of phosphorylation as well as compensatory signaling mechanisms that modulate kinase activity over time. Nevertheless, significant inhibition of STAT5 phosphorylation was detected at concentrations within the clinically relevant plasma range of CPZ (1-20 μM), underscoring the potential therapeutic applicability of our findings.
Comment 2.
The study shows differences in the effects of CPZ on signaling pathways between FLT3-ITD and FLT3-ITD/F692L cell lines, particularly in ERK activation. These inconsistencies are not fully explained, which could indicate a more complex mechanism of action than initially proposed.
Response:
We agree with the reviewer’s assessment. Our data show that CPZ induced ERK activation in FLT3-ITD cells but not significantly in FLT3-ITD/F692L cells, while AKT activation was observed in both. This suggests that resistant clones may rely on different compensatory pathways compared to sensitive cells. Previous studies have reported that the F691L mutation alters downstream signaling and contributes to resistance through AKT activation and microenvironment-mediated survival mechanisms (Scholl et al., Cells 2020; Dumas et al., Haematologica 2019). We have incorporated this mechanistic interpretation and the supporting references into the revised Discussion (line 484~488 in revision).
New references have been added as “[43,44]” to the revision. As a result, the reference numbers below have been shifted down.
[43] Scholl S, Fleischmann M, Schnetzke U, et al.
Molecular Mechanisms of Resistance to FLT3 Inhibitors in Acute Myeloid Leukemia: Ongoing
Challenges and Future Treatments.
Cells. 2020;9:2493.
[44] Dumas PY, Naudin C, Martin-Lannerée S, et al.
Hematopoietic niche drives FLT3-ITD acute myeloid leukemia resistance to quizartinib via
STAT5-and hypoxia-dependent upregulation of AXL.
Haematologica. 2019 Oct;104(10):2017-2027.
This observation suggests that FLT3-ITD/F692L cells may bypass ERK-dependent signaling and instead rely more heavily on alternative pathways such as AKT signaling. Previuos studies have reported that the analogous human F691L mutation can alter downstream signaling and promote resistance through enhanced AKT activation and microenvironment-mediated survival mechanisms[43,44].
Comment 3.
While the study examines the combination of CPZ with quizartinib and a STAT5 inhibitor, it does not explore combinations with other standard-of-care treatments for AML. This limits the translational potential of the findings.
Response:
We thank the reviewer for this important point. In the revised Discussion, we now emphasize that future studies should investigate CPZ in combination with other standard-of-care AML agents (e.g., cytarabine, daunorubicin, azacitidine, venetoclax, etc.) to better evaluate its translational potential (line 531~533 in revision).
In addition, it will be important for future studies to investigate CPZ in combination with other standard-of-care AML treatments, such as cytarabine, daunorubicin, azacitidine, and venetoclax to further assess its translational potential.
Comment 4.
CPZ is known to have multiple effects on cellular processes beyond its intended target. The study does not thoroughly address potential off-target effects that could contribute to the observed anti-leukemic activity. This limitation makes it difficult to attribute the effects solely to FLT3-ITD inhibition.
Response:
We fully agree with the reviewer’s insightful comment. CPZ is indeed known to exert multiple pleiotropic effects beyond its originally intended targets. So, we had already addressed this point in Discussion section (lines 520–526 in revision) in the original manuscript as follows:
In support of this, recent studies have demonstrated that CPZ induces G2/M cell cycle arrest [38,39], promotes apoptosis via mitochondrial and lysosomal pathways [40], modulates autophagy, increases reactive oxygen species (ROS) production [41,42], and inhibits P-glycoprotein (P-gp)-mediated drug efflux in various cancer types [43,44]. These observations suggest that CPZ exerts pleiotropic effects on cancer cells and may overcome resistance mechanisms related to ATP-competitive inhibition.
Comment 5.
The Western blot data are clear. The quantitative analysis would benefit from error bars (SEM) and statistical analysis to confirm the stated significances.
Response:
We thank the reviewer for this suggestion. As noted above, we have added quantitative densitometry data with SEM and statistical tests to Figure 3.
Comment 6.
The additive effect with the STAT5 inhibitor is interesting. The authors should clarify whether the STAT5 inhibitor was used at a concentration that fully inhibits STAT5 phosphorylation (which the blot in 5A suggests it does) and discuss why the effect is only additive and not synergistic.
Response:
We appreciate the reviewer’s careful reading. The STAT5 inhibitor was indeed used at 100 nM, a concentration known to fully inhibit STAT5 phosphorylation. Under this condition, further suppression of STAT5 signaling by CPZ could not be achieved, leading to an additive rather than synergistic effect. This observation supports the notion that CPZ exerts partial STAT5-independent activity, consistent with its pleiotropic actions on other signaling pathways such as AKT and ERK. We have clarified this explanation in the revised Discussion (line 491~495 in revision).
The additive, rather than synergistic, effect is likely because the STAT5 inhibitor was used at a concentration sufficient to fully block STAT5 phosphorylation, leaving little room for further suppression by CPZ. This outcome suggests that CPZ exerts its effect via STAT5-independent pathway(s), consistent with its pleiotropic actions.
Reviewer 2 Report
Comments and Suggestions for Authors
The manuscript is well written. My only request is that the authors include more structural information in the Introduction section. If possible, I would like them to include a figure on FLT3 structure highlighting the active site(s) and the locations of tandem repeat duplications or point mutations. Also, it would be beneficial to include a visualization of different kinase inhibitors binding to the enzyme and to correlate that with different mutations.
I would encourage the authors to begin thinking about direct binding measurements such as ITC or FRET between CPZ and its suspected targets.
Finally, there are a few minor formatting issues, e.g. "stem cell transplantation4." (line 40) and "major clinical challenge16,17." (line 360).
Author Response
Response to Reviewer 2
We thank the reviewer for the constructive feedback and encouraging comments. We have carefully addressed all the points as follows:
Comment 1.
The manuscript is well written. My only request is that the authors include more structural information in the Introduction section. If possible, I would like them to include a figure on FLT3 structure highlighting the active site(s) and the locations of tandem repeat duplications or point mutations. Also, it would be beneficial to include a visualization of different kinase inhibitors binding to the enzyme and to correlate that with different mutations.
Response:
We appreciate this valuable suggestion. In the revised Introduction, we have added a description of the FLT3 protein structure, including the extracellular, transmembrane, juxtamembrane, and kinase domains. We specifically note that ITDs commonly occur in the juxtamembrane domain, while the gatekeeper mutation F691L is located within the ATP-binding pocket of the kinase domain (line 65~74 in revision). We have also included a new figure (Supplemental Figure 1 in revision) illustrating the overall FLT3 structure, highlighting the ITD region, the F691L mutation site, and representative binding modes of type I and type II inhibitors. This addition provides readers with a clearer mechanistic context for the resistance conferred by FLT3 mutations. Supporting references have been added (Smith et al., 2012, Daver et al. 2019, Kiyoi et al. 2019, Desikan et al. 2022, Garciaz et al. 2023). As a result, the reference numbers below have been shifted down.
[18] Smith CC, Wang Q, Chin CS, et al.
Validation of ITD mutations in FLT3 as a therapeutic target in human acute myeloid leukaemia.
Nature. 2012 Apr 15;485(7397):260-3.
[19] Daver N, Schlenk RF, Russell NH, et al.
Targeting FLT3 mutations in AML: review of current knowledge and evidence.
Leukemia. 2019;33(2):299-312.
[20] Kiyoi H, Kawashima N, Ishikawa Y.
FLT3 mutations in acute myeloid leukemia: Therapeutic paradigm beyond inhibitor development
Cancer Science. 2020;111:312–322.
[21] Desikan SP, Daver N, DiNardo C, et al.
Resistance to targeted therapies: delving into FLT3 and IDH.
Blood Cancer J. 2022;12(6):91.
[22] Garciaz S, Hospital MA.
FMS-Like Tyrosine Kinase 3 Inhibitors in the Treatment of Acute Myeloid Leukemia: An Update on the Emerging Evidence and Safety Profile.
Onco Targets Ther. 2023;16:31-45.
FLT3 is a receptor tyrosine kinase consisting of an extracellular ligand-binding domain, a transmembrane region, a juxtamembrane domain, and an intracellular tyrosine kinase domain. FLT3-ITD typically occur in the juxtamembrane domain, leading to constitutive kinase activation, whereas point mutations such as F691L are located within the ATP-binding pocket of the kinase domain. These structural alterations disrupt the binding of type I and type II inhibitors and drive therapeutic resistance. To provide a visual overview, Supplemental Figure 1 illustrates the domain architecture of FLT3, highlights common ITD and point mutation sites, and depicts representative inhibitor binding modes [18-22].
Figure legend of supplementally Fig.1
Domain architecture of FLT3, highlights common ITD and TKD, and depicts representative inhibitor binding modes
FLT3 consists of five extracellular Ig-like domains, a juxtamembrane domain (JMD), two tyrosine kinase domains (TKDs), and a C-terminal region. Ligand (FL) binding induces dimerization, TKD phosphorylation, and downstream signaling essential for hematopoietic cell differentiation, proliferation, and stem cell self-renewal. In leukemic cells, FL stimulation promotes proliferation and suppresses apoptosis. Activating mutations include ITD in the JMD (20–28%) and TKD mutations such as D835 (5–10%), both leading to constitutive signaling. FLT3 inhibitors competitively bind to the ATP-binding site, thereby blocking kinase activity and inducing apoptosis. However, the conformation of the ATP-binding pocket differs between inactive and active states. Type I inhibitors bind either the ATP-binding site of constitutively active receptor (via ITD or TKD mutations) or the activation loop located between the TKDs, thereby blocking signaling. Type II inhibitors bind to a region adjacent to the ATP-binding site in the inactive kinase conformation, thereby inhibiting phosphorylation driven by ITD mutations. The F691L gatekeeper mutation arises in TKD1 and confers resistance to many FLT3 inhibitors, making it a critical therapeutic challenge.
Comment 2.
I would encourage the authors to begin thinking about direct binding measurements such as ITC or FRET between CPZ and its suspected targets.
Response:
We agree that determining whether CPZ directly binds to FLT3 or acts indirectly would provide valuable mechanistic insights. However, we consider that direct binding measurements such as isothermal titration calorimetry (ITC) and/or fluorescence resonance energy transfer (FRET) were beyond the scope of the present study. Accordingly, we have revised the Discussion to highlight these approaches as important directions for future research (line 534~540 in revision). We believe that such studies will clarify the molecular basis of CPZ’s antileukemic effects and further support its potential for clinical repurposing.
Although our findings suggest that CPZ interferes with FLT3 signaling, a direct molecular interaction between CPZ and FLT3 has not been established. Biophysical methods such as isothermal titration calorimetry (ITC) or fluorescence resonance energy transfer (FRET) assays would be valuable for determining whether CPZ binds directly to FLT3 or exerts its effects through indirect mechanisms. Future studies employing these approaches will help elucidate the molecular basis of CPZ activity and further strengthen the rationale for its clinical repurposing.
Comment 3.
Finally, there are a few minor formatting issues, e.g. "stem cell transplantation4." (line 40) and "major clinical challenge16,17." (line 360).
Response:
We thank the reviewer for pointing out these errors. All identified formatting issues have been corrected in the revised manuscript (line 43, 463 in revision).
Reviewer 3 Report
Comments and Suggestions for Authors
This study investigates the potential of chlorpromazine (CPZ), an antipsychotic drug, as a therapeutic agent in acute myeloid leukemia (AML) with FLT3-ITD mutations, including the gatekeeper F691L variant. Using Ba/F3 cell models, the authors show that CPZ suppresses proliferation of both FLT3-ITD and FLT3-ITD/F692L cells, while quizartinib (Quiz) is ineffective against the resistant F692L mutant. Mechanistically, CPZ reduces STAT5 phosphorylation and its growth-inhibitory effect is partially rescued by constitutively active STAT5, supporting STAT5 suppression as a key mechanism. The findings highlight CPZ as a promising strategy to overcome resistance to FLT3 inhibitors and suggest its potential use in combination therapies. Here are some minor revisions need to be improved.
- Abbreviations consistency – In the Introduction, ensure uniform use of “Quiz” vs “quizartinib” (some sections mix the two without first defining consistently).
- In Materials and Methods 2.1, the sentence “CPZ, and Quiz, a STAT5 inhibitor were purchased…” is misleading. Revise to “CPZ and Quiz were purchased from Sigma-Aldrich. A STAT5 inhibitor (Item No.15784) was purchased from Cayman Chemical.”
- In Section 2.2, format the listed primers with consistent spacing (some have stray spaces, e.g., “ggc cag tgt act tga t tgg aat a g”).
- Section 2.6: “Dunne’s post-hoc test” should be corrected to “Dunnett’s post hoc test.”
- Figure 2 legend: “transfecetd” should be corrected to “transfected.
- The study is entirely based on Ba/F3 cell models. The discussion should acknowledge this limitation and emphasize the need for in vivo validation of CPZ efficacy and safety in AML models, especially since CPZ is a repurposed psychiatric drug with known CNS effects.
- While the data support STAT5 suppression as one mechanism, CPZ also activated AKT/ERK pathways. The discussion briefly notes this, but it would benefit from elaborating on whether these compensatory signaling changes might limit CPZ’s therapeutic effect and how they could be addressed (e.g., by combining with AKT inhibitors).
Author Response
Response to Reviewer 3
We thank the reviewer for the positive evaluation and for identifying several useful corrections. We address each point below:
Comment 1.
Abbreviations consistency – In the Introduction, ensure uniform use of “Quiz” vs “quizartinib” (some sections mix the two without first defining consistently).
Response:
We have confirmed the text to ensure consistent use of “Quiz,” defined at first mention.
Comment 2.
In Materials and Methods 2.1, the sentence “CPZ, and Quiz, a STAT5 inhibitor were purchased…” is misleading. Revise to “CPZ and Quiz were purchased from Sigma-Aldrich. A STAT5 inhibitor (Item No.15784) was purchased from Cayman Chemical.”
Response:
We have corrected the sentence to read: “CPZ and Quiz were purchased from Sigma-Aldrich. A STAT5 inhibitor (Item No.15784) was purchased from Cayman Chemical.” (line 111~113 in revision).
Comment 3.
In Section 2.2, format the listed primers with consistent spacing (some have stray spaces, e.g., “ggc cag tgt act tga t tgg aat a g”).
Response:
The primer sequences have been reformatted for consistency and accuracy (line 125~128 in revision).
Comment 4.
- Section 2.6: “Dunne’s post-hoc test” should be corrected to “Dunnett’s post hoc test.”
Response:
“Dunne’s post-hoc test” has been corrected to “Dunnett’s post hoc test” (line 177 in revision).
Comment 5.
Figure 2 legend: “transfecetd” should be corrected to “transfected.
Response:
“Transfecetd” has been corrected to “transfected” (line 345 in revision).
Comment 6.
The study is entirely based on Ba/F3 cell models. The discussion should acknowledge this limitation and emphasize the need for in vivo validation of CPZ efficacy and safety in AML models, especially since CPZ is a repurposed psychiatric drug with known CNS effects.
Response:
We agree that reliance on Ba/F3 models is a limitation. We have revised the Discussion to acknowledge the lack of in vivo validation and emphasized the need for studies in AML animal models to evaluate CPZ’s efficacy and safety, given its CNS activity (line 540~547 in revision).
A limitation of our study is that all experiments were performed using Ba/F3 cell models. While these systems are widely used to investigate the oncogenic potential of FLT3 mutations, they do not fully recapitulate the complexity of human AML. Moreover, as CPZ is a repurposed psychiatric drug with known CNS effects, its efficacy and safety must be validated in in vivo AML models before clinical application. Such studies will be essential to establish appropriate dosing strategies that maximize antileukemic activity while minimizing neurological side effects.
Comment 7.
While the data support STAT5 suppression as one mechanism, CPZ also activated AKT/ERK pathways. The discussion briefly notes this, but it would benefit from elaborating on whether these compensatory signaling changes might limit CPZ’s therapeutic effect and how they could be addressed (e.g., by combining with AKT inhibitors).
Response:
We have elaborated in the Discussion on the potential role of compensatory AKT/ERK activation in limiting CPZ’s therapeutic effect. We also note preliminary observations (not shown) where co-treatment with an AKT inhibitor (capivasertib) enhanced CPZ’s activity, suggesting possible future combination strategies (line 509~514 in revision).
These findings indicate that CPZ-induced activation of AKT, and to a lesser extent ERK, may represent compensatory signaling that counteracts its growth-inhibitory effects. Therefore, combining CPZ with inhibitors targeting these pathways, such as AKT inhibitors, could further enhance therapeutic efficacy. Such combination approaches warrant further investigation as potential strategies to overcome compensatory signaling and optimize the clinical benefit of CPZ.
Round 2
Reviewer 1 Report
Comments and Suggestions for Authors
I appreciate the authors’ detailed and thoughtful responses to my comments. The revisions made to the manuscript, including the addition of quantitative analyses with SEM and statistical testing, clarification of mechanistic differences between FLT3-ITD and FLT3-ITD/F692L cells, expansion of the discussion on pleiotropic effects of CPZ, and consideration of its potential in combination with standard-of-care AML therapies, satisfactorily address all of my concerns. The explanations regarding the additive rather than synergistic effect with the STAT5 inhibitor are clear and consistent with the data presented.
Overall, the authors have made significant improvements that enhance the clarity, rigor, and translational relevance of the work. I believe the manuscript in its current form is suitable for publication.